# ExGS: Extreme 3D Gaussian Compression with Diffusion Priors

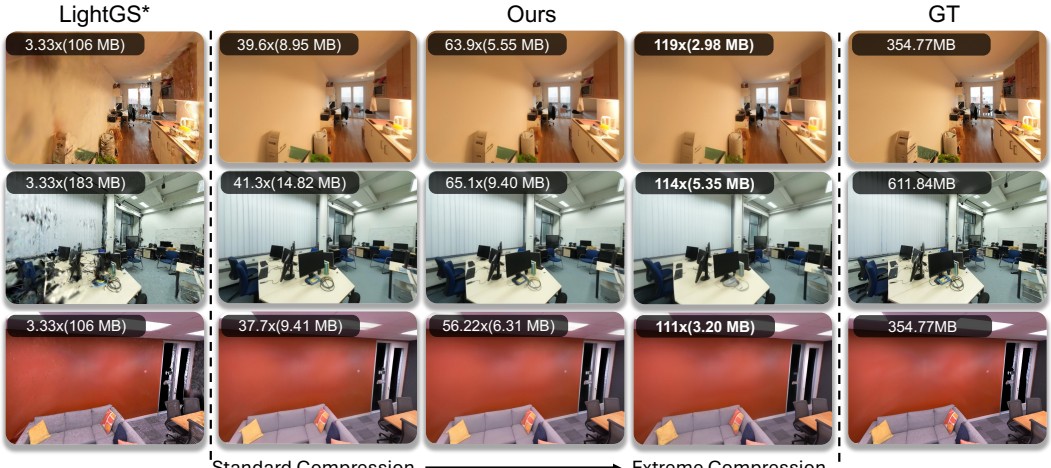

Figure 1: **Qualitative comparison under different compression ratios.** LightGS* indicates the optimized version of LightGS(Fan et al., 2024). LightGS* achieves only limited compression (around 3×) and produces severe visual artifacts. In contrast, our method realizes orders-of-magnitude higher compression (exceeding 100× in some cases) while preserving scene geometry and appearance realism. Even at aggressive compression ratios, our reconstructions remain close to ground-truth quality, demonstrating the robustness of the proposed framework.

## ABSTRACT

Neural scene representations, such as 3D Gaussian Splatting (3DGS), have enabled high-quality neural rendering; however, their large storage and transmission costs hinder deployment in resource-constrained environments. Existing compression methods either rely on costly optimization, which is slow and scene-specific, or adopt training-free pruning and quantization, which degrade rendering quality under high compression ratios. In contrast, recent data-driven approaches provide a promising direction to overcome this trade-off, enabling efficient compression while preserving high rendering quality. We introduce **ExGS**, a novel feed-forward framework that unifies **Universal Gaussian Compression** (UGC) with **GaussPainter** for **Ex**treme 3**DGS** compression. **UGC** performs re-optimization-free pruning to aggressively reduce Gaussian primitives while retaining only essential information, whereas **GaussPainter** leverages powerful diffusion priors with mask-guided refinement to restore high-quality renderings from heavily pruned Gaussian scenes. Unlike conventional inpainting, GaussPainter not only fills in missing regions but also enhances visible pixels, yielding substantial improvements in degraded renderings. To ensure practicality, it adopts a lightweight VAE and a one-step diffusion design, enabling real-time restoration. Our framework can even achieve over 100× compression (reducing a typical 354.77 MB model to about 3.31 MB) while preserving fidelity and significantly improving image quality under challenging conditions. These results highlight the central role of diffusion priors in bridging the gap between extreme compression and high-quality neural rendering.

# 1 INTRODUCTION

In recent years, 3D Gaussian Splatting (Kerbl et al., 2023) has driven rapid progress in neural rendering and novel view synthesis. 3DGS represents scenes explicitly with millions of anisotropic Gaussian primitives and achieves real-time rendering by projecting and blending these primitives in screen space. This explicit representation significantly reduces rendering costs and has quickly become a popular solution for efficient scene reconstruction and rendering. Such efficiency has enabled a wide range of practical applications, including augmented/virtual reality, mobile visualization, remote collaboration, and digital twin systems. Nevertheless, the large number of Gaussian primitives introduces heavy storage and transmission overhead, often reaching very large file sizes per scene, sometimes exceeding gigabytes, which severely limits deployment on mobile devices and bandwidth-constrained environments.

To resolve the challenging problem of overwhelming redundancy in Gaussian primitives, existing methods can be broadly categorized into two groups: optimization-based approaches (Niedermayr et al., 2024; Liu et al., 2025b) that fine-tune Gaussian parameters to reduce redundancy, and training-free approaches (Fan et al., 2024; Tian et al., 2025) that rely on quantization or heuristic pruning without additional retraining. Optimization-based methods often achieve higher fidelity, but they are computationally expensive and scene-specific; in contrast, training-free methods are lightweight and scalable but struggle under aggressive compression ratios, leading to degraded rendering quality. This trade-off between compression ratio and fidelity makes it difficult for current approaches to achieve both simultaneously.

Inspired by recent data-driven methods (Chen et al., 2025a;b), we propose **ExGS**, a novel feed-forward compression framework for 3DGS that leverages generative priors to overcome these limitations through two complementary modules: **Universal Gaussian Compression (UGC)** and **GaussPainter**. The first module performs extreme Gaussian compression without re-optimization, retaining only the most essential information and maximally leveraging the prior knowledge of generative models. The second module, GaussPainter, exploits the strong priors of diffusion models and a mask-guided refinement strategy to restore high-quality renderings, even from heavily pruned and incomplete Gaussian scenes.

By combining **UGC** and **GaussPainter**, our framework achieves orders of magnitude compression, often exceeding $100\times$, while still enabling real-time rendering and high-quality restoration. As illustrated in Fig. 1, prior approaches tend to degrade noticeably once the compression ratio becomes aggressive, showing blurred textures and structural distortions. In contrast, our method maintains geometric fidelity and perceptual realism even under much stronger compression, demonstrating its robustness and clear advantage over existing baselines. **Our contributions are threefold:**

(1) We demonstrate, for the first time, that a data-driven paradigm can drive generative 3DGS compression, moving beyond heuristic pruning or optimization-based tuning by leveraging powerful generative priors for faithful restoration under extreme compression.

(2) We propose two complementary modules: **UGC** for re-optimization-free pruning and **GaussPainter** for diffusion-based refinement, and demonstrate that their synergy enables compact representations while preserving high visual fidelity.

(3) Extensive experiments on public datasets demonstrate that **ExGS** achieves state-of-the-art performance, maintaining real-time rendering and robust quality even under a storage size compression ratio of up to $100\times$.

# 2 RELATED WORK

## 2.1 EFFICIENT NEURAL RENDERING.

3D Gaussian Splatting (3DGS) has rapidly emerged as a powerful representation for real-time scene rendering, yet its efficiency is often constrained by the large number of Gaussian primitives. To improve scalability, prior works have explored compression and pruning strategies. Codebook-based methods (Navaneet et al., 2024; Lee et al., 2024; Fan et al., 2024) reduce the storage footprint by quantizing Gaussian parameters, while pruning approaches (Fan et al., 2024; Tian et al., 2025) directly eliminate redundant primitives to achieve compact representations. Beyond pure reduction,

densification techniques such as Mini-Splatting (Fang & Wang, 2024) and Taming-3DGS (Mallick et al., 2024) regenerate new primitives in a spatially efficient manner, whereas Scaffold-GS (Lu et al., 2024) anchors Gaussians on structured grids for better distribution.

Although these approaches have demonstrated strong effectiveness, they often rely on retraining or fixed pruning ratios, limiting flexibility under different compression requirements. In contrast, our method improves upon the LightGS (Fan et al., 2024) scoring mechanism and further integrates generative priors, enabling controllable compression while preserving rendering quality even at extreme ratios.

## 2.2 Diffusion Priors for 3D Reconstruction.

Diffusion models have recently shown strong potential in addressing degradation in 3D reconstruction tasks. Traditional dense methods, such as NeRF-based pipelines, require abundant multi-view inputs, while sparse-view methods (Wang et al., 2023; Chen et al., 2023b) attempt to compensate with structural priors but still suffer from incomplete reconstructions. Diffusion models extend this line of work by providing powerful generative priors that can restore missing details and refine degraded regions.

Several recent works integrate diffusion with Gaussian-based or NeRF-based reconstructions. For instance, Difix3D+ (Wu et al., 2025) leverages diffusion to denoise target views using reference information, yielding improved fidelity. Generative Gaussian Splatting (GGS) (Schwarz et al., 2025) integrates video diffusion priors with Gaussian splatting to enhance view consistency, and GSFixer (Yin et al., 2025) employs reference-guided diffusion to correct artifacts in sparse inputs. Similarly, GSD (Mu et al., 2024) combines Gaussian splatting and diffusion priors for single-view 3D reconstruction, while Single-Stage Diffusion NeRF (SSDNeRF) (Chen et al., 2023a) jointly learns diffusion priors and neural fields from sparse inputs. Diffusion has also been explored for human reconstruction, e.g., DiHuR (Chen et al., 2024), which uses diffusion priors to enhance geometry and texture fidelity under sparse or degraded observations.

Different from these approaches, our method directly applies diffusion priors to compressed Gaussian renderings, without requiring extra views or heavy supervision. By combining mask-guided conditioning with efficient one-step diffusion and latent alignment, it is capable of handling diverse degradation sources such as sparse inputs, severe pruning, and structural inconsistencies, thereby producing more consistent and high-quality reconstructions.

## 3 Preliminaries

### 3.1 3D Gaussian Splatting

3D Gaussian Splatting Kerbl et al. (2023) explicitly represents 3D scenes with a collection of anisotropic Gaussian primitives. Each Gaussian is parameterized by its mean position $\mu \in \mathbb{R}^3$, covariance matrix $\Sigma \in \mathbb{R}^{3\times3}$, opacity $\sigma \in (0, 1]$, and color (or feature) $\mathbf{c} \in [0, 1]^3$. During rendering, the color $C$ of a pixel is obtained by blending $N$ ordered Gaussians that overlap with the pixel using alpha blending:

$$C = \sum_{i\in N} \mathbf{c}_i \alpha_i \prod_{j=1}^{i-1}(1-\alpha_j), \tag{1}$$

where $\alpha_i$ denotes the opacity contribution of the $i$-th Gaussian at the pixel, typically computed by evaluating the projected 2D elliptical footprint defined by $\Sigma_i$ and scaling with its opacity $\sigma_i$. This design enables differentiable and efficient rendering in real time, making 3DGS a strong alternative to volumetric ray marching methods.

### 3.2 Diffusion Models

Diffusion models Ho et al. (2020); Song et al. (2020) generate data by adding Gaussian noise in a forward Markov chain and learning to reverse it. In the forward process, noise is added step by step:

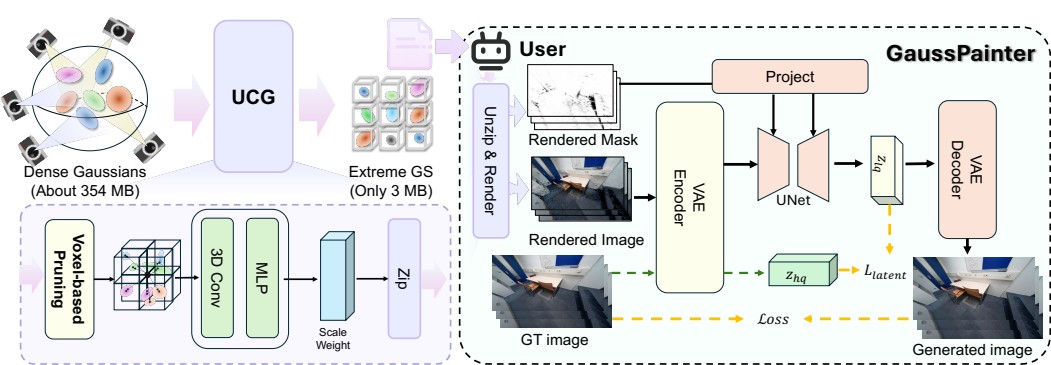

Figure 2: **The overall pipeline of ExGS.** Dense Gaussians are compressed into a compact 3 MB representation via pruning and lightweight modules, then decompressed and processed by the VAE and UNet to reconstruct high-quality images. This design enables efficient storage, fast transmission, and high-fidelity rendering.The dashed arrows in the figure indicate components that are used only during training.

$$q(\mathbf{z}_{1:T} \mid \mathbf{z}_0) = \prod_{t=1}^{T} \mathcal{N}\left(\mathbf{z}_t; \sqrt{1-\beta_t}\,\mathbf{z}_{t-1},\, \beta_t \mathbf{I}\right), \tag{2}$$

which has the closed form

$$q(\mathbf{z}_t \mid \mathbf{z}_0) = \mathcal{N}\left(\mathbf{z}_t; \sqrt{\bar{\alpha}_t}\,\mathbf{z}_0,\, (1-\bar{\alpha}_t)\mathbf{I}\right), \tag{3}$$

where $\alpha_t = 1 - \beta_t$ and $\bar{\alpha}_t = \prod_{i=1}^{t} \alpha_i$.

**Reverse process.** Sampling starts from Gaussian noise $\mathbf{z}_T \sim \mathcal{N}(0, I)$ and reconstructs $\mathbf{z}_0$ via

$$q(\mathbf{z}_{t-1} \mid \mathbf{z}_t, \mathbf{z}_0) = \mathcal{N}\left(\mathbf{z}_{t-1}; \mu_t(\mathbf{z}_t, \mathbf{z}_0),\, \sigma_t^2 \mathbf{I}\right), \tag{4}$$

with mean

$$\mu_t(\mathbf{z}_t, \mathbf{z}_0) = \frac{1}{\sqrt{\alpha_t}}\left(\mathbf{z}_t - \frac{1-\alpha_t}{\sqrt{1-\bar{\alpha}_t}}\,\boldsymbol{\epsilon}\right), \tag{5}$$

where $\boldsymbol{\epsilon}$ is the injected noise. A neural network $\boldsymbol{\epsilon}_\theta(\mathbf{z}_t, t, \mathbf{c})$ predicts $\boldsymbol{\epsilon}$, optionally conditioned on $\mathbf{c}$.

The training objective is the noise-prediction loss:

$$\mathcal{L}_{\text{DM}} = \mathbb{E}_{\mathbf{z}_0, t, \boldsymbol{\epsilon}} \left\|\boldsymbol{\epsilon} - \boldsymbol{\epsilon}_\theta\left(\sqrt{\bar{\alpha}_t}\mathbf{z}_0 + \sqrt{1-\bar{\alpha}_t}\,\boldsymbol{\epsilon}, t\right)\right\|_2^2. \tag{6}$$

## 4 METHOD

In this section, we introduce **ExGS**, a novel feed-forward Gaussian compression framework that is designed to simultaneously improve the rendering quality of incomplete Gaussian scenes and alleviate the transmission overhead of Gaussian representations. Unlike existing approaches that either focus on scene fidelity or transmission efficiency, our method unifies both objectives in a single architecture. As illustrated in Fig. 2, the framework consists of two key components: the **Universal Gaussian Compression** module (Sec. 4.1) that compactly encodes Gaussian scenes, and the **GaussPainter** module (Sec. 4.2) that reconstructs high-quality renderings at nearly real-time speed with extremely low transmission cost.

### 4.1 UNIVERSAL GAUSSIAN COMPRESSION

With 3DGS increasingly adopted as a standalone 3D representation, practical pipelines typically expose only the 3DGS scene file, while ground-truth imagery remains inaccessible. Under these

constraints, there is a strong need for fast and controllable compression of Gaussian representations. To address this, we propose Universal Gaussian Compression, a module that flexibly controls the number of Gaussian primitives while preserving geometric structure and rendering details. This reduces storage and transmission costs while maintaining fidelity for downstream tasks.

Our compression method consists of two components: (i) a global significance score for evaluating the importance of each Gaussian, and (ii) a voxel-based pruning strategy that enforces spatial compactness while supplementing under-represented regions.

**Global Significance Score.** To measure the contribution of each Gaussian primitive, we define a global significance score $GS_j$ considering its visibility, opacity, and ray transmittance (Fan et al., 2024):

$$GS_j = \sum_{i=1}^{MHW} \mathbf{1}\big(G(\mathbf{X}_j), r_i\big) \cdot \sigma_j \cdot T_j, \tag{7}$$

where $i$ indexes all pixels across $M$ views with image resolution $H \times W$. The indicator function $\mathbf{1}(\cdot)$ returns 1 if the $j$-th Gaussian $G(\mathbf{X}_j)$ intersects with the camera ray $r_i$, and 0 otherwise. $\sigma_j$ denotes the opacity of the $j$-th Gaussian, and $T_j$ represents the accumulated transmittance of the ray before reaching $G(\mathbf{X}_j)$, defined as:

$$T_j = \prod_{i=1}^{j-1}(1 - \sigma_i). \tag{8}$$

We observe that the importance score is often influenced by the viewing direction and the geometric position of Gaussians, as illustrated in Fig. 3. While it effectively preserves color details in many cases, its performance is less reliable in sparsely textured regions. To overcome this limitation, we introduce a voxel-based pruning strategy that complements the scoring function. With the same number of Gaussians, this sampling approach achieves more complete preservation of both global geometric structures and fine-grained local details.

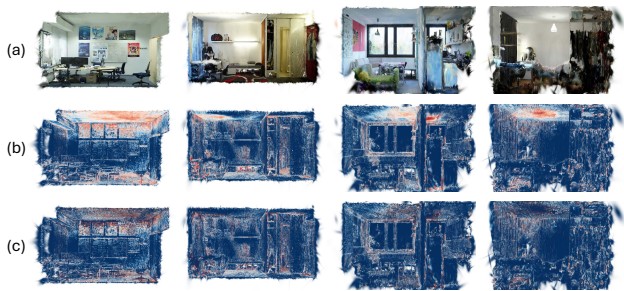

Figure 3: **Visualization of global significance scoring and voxel-based correction.** (a) shows rendered views of Gaussian scenes. (b) presents heatmaps of the raw score, which emphasize textures but fail in sparse regions. (c) shows the corrected distribution after voxel-based pruning, yielding more uniform coverage and improved preservation of structure and details.

**Voxel-based Pruning.** The significance score does not ensure spatial uniformity: dense regions may retain redundancy, while sparse ones may lose coverage. We partition the space into voxels of size $v$, group Gaussians by voxel, and ensure each voxel with sufficient points contributes at least one selected Gaussian. Within a voxel $\mathcal{V}$ containing Gaussians $\{g_j\}$ and scores $s_j$, we select

$$\mathcal{S}_{\mathcal{V}} = \arg \max_{\substack{\mathcal{S} \subset \mathcal{V} \\ |\mathcal{S}|=k}} \sum_{g_j \in \mathcal{S}} s_j, \tag{9}$$

where $k = \lfloor |\mathcal{V}| \cdot \rho \rfloor$. The final set combines globally selected Gaussians with voxel-based supplements, balancing global importance and spatial coverage.

**Adaptive Amplification.** Even after pruning, some regions may remain sparse, leaving holes that degrade rendering. To mitigate this, we predict an amplification factor for retained Gaussians:

$$\hat{s}_j = f_\theta(g_j), \tag{10}$$

where $f_\theta(\cdot)$ is a neural predictor. The factor $\hat{s}_j$ up-weights contributions of selected Gaussians, enriching spatial distribution and preventing unrealistic artifacts during generation.

Finally, to achieve extreme compression, we omit the storage of high-order spherical harmonics since GaussPainter does not rely on them. All remaining parameters are stored in `float16` precision and further subjected to LZMA-based lossless compression. With these optimizations, the representation can be reduced to only about 3 MB on Replica (Straub et al., 2019), achieving compression ratios of over $100\times$ while remaining practical for large-scale deployment and real-time streaming (see Appendix A.1 for detailed compression scheme).

## 4.2 GAUSSPAINTER: EFFICIENT MASK-GUIDED DIFFUSION FOR GAUSSIAN SCENES

The generative module is designed to leverage the strong priors of diffusion models to address the information loss introduced by aggressive Gaussian compression. While diffusion excels at synthesizing realistic content, it often suffers from two limitations in this setting: (i) hallucination of non-existent structures, and (ii) prohibitive time costs due to iterative sampling.

**Mitigating Hallucinations.** To mitigate hallucination in compressed Gaussian renderings, we employ two complementary strategies: **latent supervision** and **mask guidance**. For **latent supervision**, relying only on pixel-level losses fails to reliably separate regions that should be preserved from those requiring completion, as corrupted or blackened areas introduced by Gaussian compression are often misread as valid content (see Fig. 4). Inspired by deblurring models (Liu et al., 2025a), we therefore impose supervision directly in the latent space: the ground-truth image $y$ and the degraded rendering $x$ are encoded by the VAE as $z_{\text{hq}}=E(y)$ and $z_{\text{lq}}=E(x)$, and we minimize

$$\mathcal{L}_{\text{latent}} = \|z_{\text{lq}} - z_{\text{hq}}\|_2^2,$$

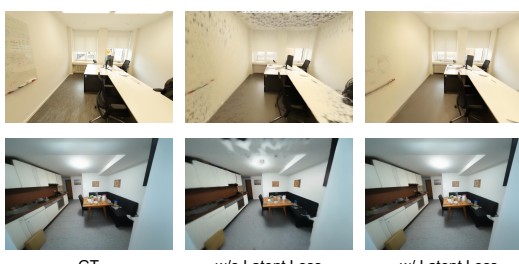

GT   w/o Latent Loss   w/ Latent Loss

Figure 4: **Comparison of results without and with latent supervision.** Latent loss effectively removes artifacts and improves reconstruction fidelity.

encouraging the degraded latent to approach the manifold of its high-quality counterpart and helping the model decide what to preserve versus plausibly complete.

While this latent alignment improves structural completion, large holes or low-texture regions may still exhibit color drift (see Fig. 6). To address this, we integrate **mask guidance**. A visibility mask derived from accumulated opacity in 3DGS rendering (Eq. 1) is normalized, flattened, globally pooled, and projected into the text-conditioning embedding space. The resulting mask embedding is then added element-wise to the caption features, which down-weights already reliable areas and steers the denoiser toward pixels likely to be missing or corrupted. This improves boundary handling around holes and reduces the tendency to propagate artifacts originating from compression.

At inference, the latent of the incomplete rendering is denoised at a fixed timestep $t=199$ in a single forward pass of the U-Net, where the injected mask embedding supplies spatial cues for where to inpaint versus preserve. This lightweight integration adds negligible overhead yet substantially reduces hallucinations and stabilizes color and illumination. Consistent with our overall design of directly applying diffusion priors to compressed Gaussian renderings without extra views or heavy supervision, combining mask-guided conditioning with efficient one-step diffusion and latent alignment enables GaussPainter to handle diverse degradation sources such as severe pruning, sparse inputs, and structural inconsistencies, thereby producing reconstructions that are structurally complete and visually faithful.

**Real-Time Rendering.** A key advantage of our framework is its ability to operate in near real-time, making it suitable for practical deployment in interactive applications. To achieve this, we replace the standard VAE with TAESD (Bohan, 2023), a lightweight autoencoder designed for fast encoding and decoding while maintaining visual fidelity. In addition, we adopt a one-step diffusion schedule, where the denoiser directly predicts the clean latent in a single forward pass at a fixed timestep, bypassing the expensive iterative sampling typical of diffusion models. Together, these two design choices drastically reduce inference latency, enabling high-quality restoration of compressed Gaussian renderings at interactive frame rates.

Figure 5: **Qualitative results at a 10% compression ratio on ScanNet++, Mip-NeRF360 and Replica**. Visual comparisons show that our method reconstructs scenes with higher fidelity and completeness compared to baseline approaches. All generative methods take as input the same UGC-enhanced rendered images for fair comparison.

## 5 EXPERIMENT

### 5.1 EXPERIMENTAL SETUPS

**Evaluation Protocol.** We evaluate our method on indoor scenes using Replica(Straub et al., 2019) and 50 test scenes from ScanNet++ Yeshwanth et al. (2023), as well as on outdoor scenes with novel-view synthesis from Mip-NeRF360 Barron et al. (2021). Across multiple compression ratios, we report LPIPS, SSIM, and PSNR, and additionally record the corresponding storage size at each ratio. In addition, we evaluate the inference speed of generative methods on an NVIDIA A100 GPU to demonstrate the efficiency of our approach. Each model is first warmed up with two iterations, after which we perform 10 forward passes at a resolution of $512^2$, recording the average runtime and the corresponding standard deviation in Tab.4c.

**Baselines.** We compare our framework with both compression-based and generative methods. For compression, we include LightGaussian (LightGS) (Fan et al., 2024) and FlexGaussian (Tian et al., 2025), both training-free approaches that do not require ground-truth images. Since spherical harmonics (SH) contribute substantially to storage size, we remove high-order SH components in our method for more extreme compression; accordingly, we report results for LightGS both with and without SH. In addition, we evaluate the original optimization-based LightGS by running its official optimization routine for 1,000 steps on the random views. For generative baselines, we evaluate PrefPaint (Liu et al., 2024) and the inpainting variant of Stable Diffusion 2 (Rombach et al., 2022). We also compare with Difix3D (Wu et al., 2025), which applies single-step diffusion to suppress residual artifacts. This setup ensures a fair comparison across compression-only, optimization-based, and generative approaches.

**Implementation Details.** To preserve fine details under heavy compression while maintaining efficiency, we enhance the VAE decoder with lightweight skip adapters and integrate LoRA modules for parameter-efficient adaptation. During training, the full UNet denoiser is optimized, while only the low-rank adapters on the VAE are updated, balancing flexibility and stability. The overall loss is

$$\mathcal{L} = \mathcal{L}_{L2} + \lambda_p \, \mathcal{L}_{LPIPS} + \lambda_{lat} \, \mathcal{L}_{latent},$$

Table 1: **Quantitative results on ScanNet++ under different compression ratios of Gaussian counts, reporting LPIPS↓, SSIM↑, PSNR↑, compression ratio relative to the source file (404.73 MB).** LightGS* indicates the optimized version of LightGS. **Bold** indicates the best performance, and underline denotes the second-best.

| Method | 0.1 | | | | 0.2 | | | | 0.3 | | | |
|---|---|---|---|---|---|---|---|---|---|---|---|---|
| | LPIPS↓ | SSIM↑ | PSNR↑ | Ratio (Size, MB) | LPIPS↓ | SSIM↑ | PSNR↑ | Ratio (Size, MB) | LPIPS↓ | SSIM↑ | PSNR↑ | Ratio (Size, MB) |
| LightGS (w/o SH) | 0.4583 | 0.6130 | 15.56 | 36.5× (11.10) | 0.3336 | 0.7577 | 19.71 | 18.2× (22.19) | 0.2667 | 0.8270 | 22.45 | 12.2× (33.29) |
| LightGS (with SH) | 0.4539 | 0.6186 | 15.64 | 10.0× (40.47) | 0.3214 | 0.7750 | 20.20 | 5.0× (80.95) | 0.2498 | 0.8510 | 23.45 | 3.3× (121.42) |
| LightGS* | 0.4062 | 0.6842 | 18.87 | 10.0× (40.47) | 0.3374 | 0.7393 | 20.59 | 5.0× (80.93) | 0.2962 | 0.7774 | 21.78 | 3.3× (121.41) |
| FlexGaussian | 0.4887 | 0.6833 | 15.62 | 60.3× (6.71) | 0.4300 | 0.7899 | 20.09 | 31.3× (12.92) | 0.3704 | 0.8408 | 23.23 | 21.2× (19.12) |
| Difix3D | 0.4449 | 0.6037 | 15.60 | – | 0.3188 | 0.7528 | 19.67 | – | 0.2535 | 0.8228 | 22.33 | – |
| PrefPaint | 0.5518 | 0.5263 | 13.91 | – | 0.4547 | 0.6323 | 15.28 | – | 0.4086 | 0.6837 | 16.76 | – |
| SD2-Inpaint | 0.5893 | 0.4691 | 12.64 | – | 0.4988 | 0.5691 | 13.63 | – | 0.4583 | 0.6130 | 14.07 | – |
| **Ours** | **0.2321** | **0.8122** | **22.93** | **109.1× (3.71)** | **0.1948** | **0.8387** | **24.29** | **60.4× (6.70)** | **0.1718** | **0.8603** | **25.25** | **37.9× (10.68)** |

Table 2: **Quantitative comparison of different methods under various compression ratios of Gaussian counts on Replica.** We report LPIPS↓, SSIM↑, PSNR↑, and the compression ratio relative to the source file (354.77 MB). LightGS* indicates the optimized version of LightGS. **Bold** indicates the best performance, and underline denotes the second-best.

| Method | 0.1 | | | | 0.2 | | | | 0.3 | | | |
|---|---|---|---|---|---|---|---|---|---|---|---|---|
| | LPIPS↓ | SSIM↑ | PSNR↑ | Ratio (Size, MB) | LPIPS↓ | SSIM↑ | PSNR↑ | Ratio (Size, MB) | LPIPS↓ | SSIM↑ | PSNR↑ | Ratio (Size, MB) |
| LightGS (w/o SH) | 0.4158 | 0.7000 | 18.51 | 36.5× (9.73) | 0.2645 | 0.8402 | 23.07 | 18.2× (19.46) | 0.2130 | 0.8812 | 25.44 | 12.2× (29.18) |
| LightGS (with SH) | 0.3871 | 0.7150 | 18.93 | 10.0× (35.48) | 0.2393 | 0.8564 | 23.94 | 5.0× (70.95) | 0.1950 | 0.8969 | 26.58 | 3.3× (106.43) |
| LightGS* | 0.3242 | 0.8122 | 23.79 | 10.0× (35.47) | 0.2843 | 0.8421 | 25.34 | 5.0× (70.93) | 0.2669 | 0.8576 | 25.95 | 3.3× (106.43) |
| FlexGaussian | 0.4451 | 0.7237 | 18.91 | 59.0× (6.01) | 0.3496 | 0.8200 | 23.76 | 30.8× (11.52) | 0.3125 | 0.8510 | 26.21 | 20.8× (17.03) |
| Difix3D | 0.4264 | 0.6785 | 17.94 | – | 0.2821 | 0.8210 | 22.30 | – | 0.2358 | 0.8615 | 24.65 | – |
| PrefPaint | 0.5201 | 0.5701 | 13.76 | – | 0.4112 | 0.6688 | 15.43 | – | 0.3776 | 0.6928 | 16.02 | – |
| SD2-Inpaint | 0.5451 | 0.5299 | 12.74 | – | 0.4404 | 0.6233 | 14.05 | – | 0.4073 | 0.6482 | 14.50 | – |
| Ours | **0.2197** | **0.8298** | **24.32** | **107.2× (3.31)** | **0.1652** | **0.8717** | **26.81** | **53.3× (6.65)** | **0.1478** | 0.8844 | **27.22** | **35.9× (9.89)** |

where $\lambda_p$ and $\lambda_{\text{lat}}$ are weighting factors. *UGC* and *GaussPainter* are trained jointly under this unified objective to ensure consistency. Further implementation details are provided in Appendix A.2.

## 5.2 COMPARISONS

**Comparison on Indoor Datasets.** In the indoor datasets ScanNet++(Yeshwanth et al., 2023) and Replica(Straub et al., 2019), we compare different compression and generation methods under three compression ratios, where the ratios refer to the proportion of Gaussians retained. As shown in Tab. 1, our method achieves the best overall performance across all settings. On ScanNet++, at the most aggressive ratio (0.1) our approach delivers significant gains in perceptual and structural quality while reaching over $100\times$ compression, clearly surpassing other methods, with a PSNR improvement of about $4\,\text{dB}$ over the second-best baseline. Similar improvements are observed at ratios 0.2 and 0.3, where our method further enhances fidelity while maintaining fast inference speed. On Replica, our framework consistently outperforms alternatives, sustaining PSNR above $26\,\text{dB}$ and SSIM around 0.88 even at higher compression levels. These results confirm that our approach not only reduces artifacts and structural distortions but also preserves fine details and scene fidelity under extreme compression.

**Comparison on Outdoor Datasets.** Tab. 3 reports results on the Mip-NeRF360 (Barron et al., 2021) benchmark. Our method again surpasses both compression-based and generative baselines across all ratios. Even at an aggressive ratio of 0.03, our method significantly outperforms the second-best approach, achieving a $3.76\,\text{dB}$ gain in PSNR and a 0.06 reduction in LPIPS, while still reaching over $300\times$ compression. At moderate ratios (0.05 and 0.1), the framework maintains superior perceptual metrics with much lower distortion, highlighting its robustness to extreme compression.

## 5.3 ABLATION STUDIES

**Effect of Pipeline Modules.** To evaluate the impact of the compression stage, we conduct ablations using only *Global Significance Score pruning* as the baseline. As shown in Tab. 4a, adding UGC on top of this baseline improves structural similarity and reconstruction fidelity, while further integrating GaussPainter enhances perceptual quality. When the two modules are combined, the system achieves the best overall performance, reducing LPIPS to 0.1948 and increasing SSIM/PSNR

Table 3: **Quantitative comparison of novel view synthesis on Mip-NeRF360**. We report LPIPS↓, SSIM↑, PSNR↑, and compression ratio relative to the source file (658.91MB). LightGS* indicates the optimized version of LightGS. **Bold** indicates the best performance, and underline denotes the second-best.

| Method | 0.03 | | | | 0.05 | | | | 0.1 | | | |
|---|---|---|---|---|---|---|---|---|---|---|---|---|
| | LPIPS↓ | SSIM↑ | PSNR↑ | Ratio (Size, MB) | LPIPS↓ | SSIM↑ | PSNR↑ | Ratio (Size, MB) | LPIPS↓ | SSIM↑ | PSNR↑ | Ratio (Size, MB) |
| LightGS (w/o SH) | 0.5868 | 0.4481 | 15.71 | 121.6× (5.42) | 0.5313 | 0.5072 | 16.72 | 72.9× (9.03) | 0.4585 | 0.5936 | 17.73 | 36.5× (18.07) |
| LightGS (with SH) | 0.5802 | 0.4605 | 15.76 | 33.3× (19.77) | 0.5224 | 0.5253 | 16.83 | 20.0× (32.95) | 0.4466 | **0.6187** | 17.99 | 10.0× (65.89) |
| LightGS* | 0.6840 | 0.4104 | 15.40 | 33.3× (19.77) | 0.6286 | 0.4487 | 15.95 | 20.0× (32.95) | 0.5491 | 0.5134 | 16.82 | 10.0× (65.89) |
| FlexGaussian | 0.5626 | 0.4522 | 14.81 | 189.9× (3.47) | 0.5182 | 0.5035 | 15.91 | 120.0× (5.49) | 0.4464 | 0.5968 | 17.97 | 62.27× (10.58) |
| Difix3D | 0.5217 | 0.4402 | 15.94 | – | 0.4690 | 0.4973 | 16.97 | – | 0.3999 | 0.5885 | 18.74 | – |
| PrefPaint | 0.6574 | 0.3619 | 14.27 | – | 0.6081 | 0.3983 | 14.84 | – | 0.5670 | 0.4135 | 14.73 | – |
| SD2-Inpaint | 0.6623 | 0.3428 | 12.95 | – | 0.6248 | 0.3716 | 13.46 | – | 0.5851 | 0.3930 | 13.39 | – |
| **Ours** | **0.4595** | **0.4832** | **19.70** | **352.3× (1.87)** | **0.3934** | **0.5375** | **21.01** | **227.2× (2.90)** | **0.3303** | 0.6054 | **22.13** | **102.15× (6.45)** |

Table 4: Ablation studies of the proposed framework on ScanNet++ (Yeshwanth et al., 2023). (a) Effect of UGC and GaussPainter. (b) Effect of mask guidance and latent supervision. (c) Inference time comparison on an NVIDIA A100 GPU.

(a) Effect of pipeline components

| UGC | GaussPainter | LPIPS↓ | SSIM↑ | PSNR↑ |
|---|---|---|---|---|
| × | × | 0.3336 | 0.7577 | 19.71 |
| ✓ | × | 0.2985 | 0.7794 | 21.73 |
| × | ✓ | 0.2059 | 0.8336 | 23.24 |
| ✓ | ✓ | **0.1948** | **0.8387** | **24.29** |

(b) Effect of design choices

| Method | LPIPS↓ | SSIM↑ | PSNR↑ |
|---|---|---|---|
| Baseline | 0.3464 | 0.7480 | 21.05 |
| + Mask guidance | 0.3305 | 0.7676 | 21.73 |
| + Latent supervision | **0.2321** | **0.8122** | **22.93** |

(c) Inference time (A100)

| Method | Time (ms) |
|---|---|
| Difix3D | $230.36 \pm 16.47$ |
| PrefPaint | $2724.02 \pm 24.32$ |
| SD2-Inpaint | $2683.14 \pm 61.76$ |
| **Ours** | **$65.85 \pm 0.63$** |

to 0.8387 and 24.29 dB. This experiment is conducted on ScanNet++ (Yeshwanth et al., 2023) with a 20% pruning ratio.

**Effect of Generation.** Since GaussPainter is the core generation module, we further analyze its internal components: *mask guidance* and *latent supervision* (Tab. 4b). This ablation is carried out on the ScanNet++ test set under a 10% compression ratio, where UGC-compressed representations are provided as input to GaussPainter. Incorporating mask guidance reduces LPIPS from 0.3464 to 0.3305 and raises SSIM/PSNR to 0.7676 and 21.73 dB, showing that structural priors help the model focus on relevant regions. Adding latent supervision yields the most significant gains, with LPIPS dropping to 0.2321, SSIM increasing to 0.8122, and PSNR reaching 22.93 dB, demonstrating that latent-level consistency provides a stronger learning signal and substantially improves perceptual fidelity.

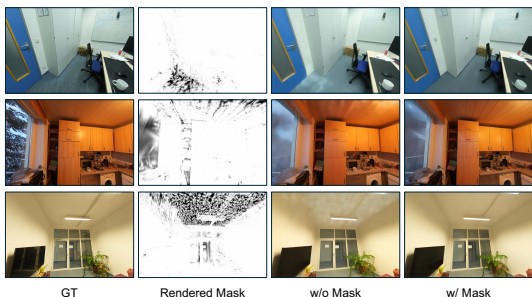

GT  Rendered Mask  w/o Mask  w/ Mask

Figure 6: Comparison of diffusion generation with and without mask guidance. From left to right: ground-truth image, rendered opacity mask from 3DGS blending, generation without mask guidance, and generation with mask guidance. The mask effectively constrains the diffusion model to complete missing regions while preserving visible structures, leading to more faithful and realistic reconstructions.

## 6 CONCLUSION

We proposed ExGS, a feed-forward Gaussian compression framework that unifies compact representation and generative refinement. Through the integration of UGC and GaussPainter, our method achieves extreme compression while maintaining high rendering fidelity. Experiments on both indoor and outdoor datasets demonstrate that ExGS realizes compression ratios of over two orders of magnitude with efficient inference, highlighting its potential for practical deployment in large-scale and real-time 3D applications.

ETHICS STATEMENT

This work aims to advance research in 3D scene compression and generative reconstruction. It does not involve human subjects, personally identifiable information, or sensitive data. The datasets used are publicly released under academic licenses and widely adopted in prior research. We follow the terms of use of these datasets and ensure that our experiments respect data privacy and integrity. We believe the research poses no foreseeable ethical risks or societal harms.

REPRODUCIBILITY STATEMENT

We have made every effort to ensure the reproducibility of our results. All datasets used in our experiments are publicly available, and preprocessing details are provided in the appendix. Model configurations, training settings, and evaluation protocols are described in the main text and supplementary material. Results are reported across multiple datasets and compression ratios to demonstrate robustness and consistency. These descriptions should enable independent researchers to reproduce our findings.

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

# A APPENDIX

## A.1 LOSSLESS COMPRESSION WITH LZMA

To reduce storage overhead while preserving numerical fidelity, we employ the Lempel–Ziv–Markov chain Algorithm (LZMA) for lossless compression. LZMA extends the classical LZ77 scheme with two key components: (i) a large sliding dictionary that replaces recurring byte sequences with back-references, thereby exploiting redundancy across long ranges; and (ii) probability-based range coding, a form of arithmetic coding, which assigns shorter codes to frequent symbols and longer codes to rare ones. Together, dictionary substitution and probabilistic modeling enable high compression ratios without discarding any information. Importantly, LZMA is strictly lossless—every byte of the original array can be reconstructed exactly after decompression, ensuring that no numerical precision is sacrificed in subsequent experiments.

## A.2 TRAINING DETAILS

All experiments are conducted on NVIDIA A100 GPUs with mixed-precision training. We train for up to 10,000 steps with a learning rate of $2 \times 10^{-5}$. The backbone components—tokenizer, text encoder, VAE, and UNet—are initialized from the pre-trained `sd-turbo` model (Rombach et al., 2022; Radford et al., 2021). We adopt a one-step diffusion scheduler implemented via `DDPMScheduler` and apply mask conditioning to guide the denoising process.

During training, *UGC* and *GaussPainter* are jointly optimized with a per-GPU batch size of 1. Each forward pass processes a single scene: Gaussian primitives are compressed by *UGC*, and a randomly sampled view is rendered into an image that *GaussPainter* uses for reconstruction.

The variational autoencoder (VAE) is augmented with LoRA (rank 4). LoRA adapters are inserted into all `Conv2d` layers of the VAE decoder, with Gaussian-initialized LoRA weights to ensure stable optimization.

## A.3 ADDITIONAL EXPERIMENTAL RESULTS

This section complements the main paper with results under additional compression ratios, extended qualitative visualizations, and further ablations.

**Indoor benchmarks at 20% and 30% ratios.** Figures 7 and 8 present qualitative comparisons on **ScanNet++** and **Replica** at pruning ratios $r=0.20$ and $r=0.30$, respectively. For each scene, we show the input rendering after compression, our reconstruction, and zoom-in crops. These visualizations corroborate the quantitative trends reported in the main paper: as the pruning ratio increases,

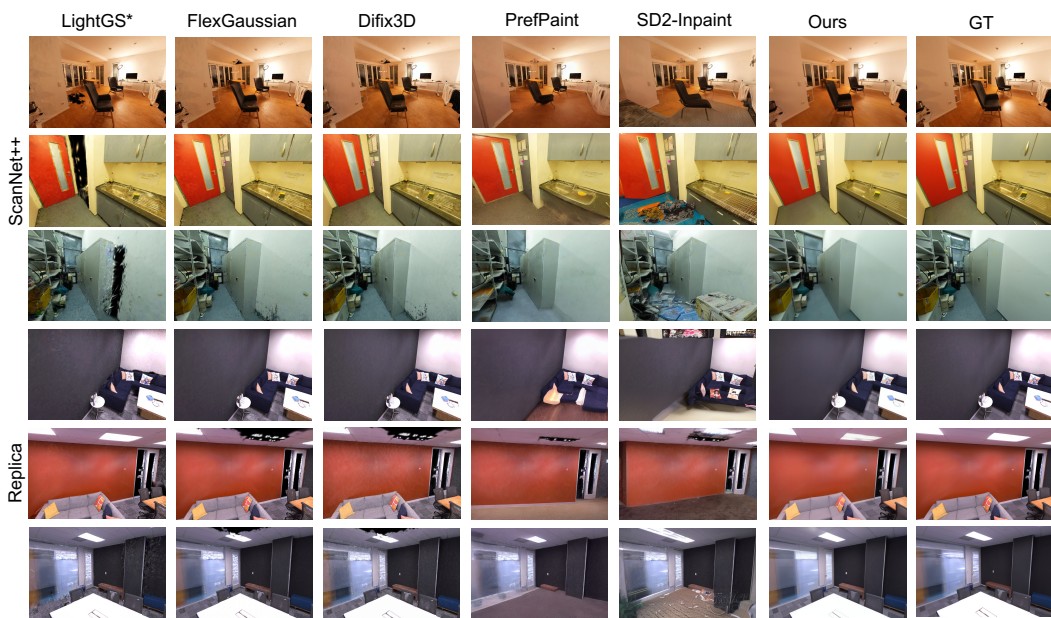

Figure 7: **Indoor results at** $r$=0.20 **(20% pruning).** Qualitative comparisons on ScanNet++ and Replica. Here $r$ denotes the fraction of Gaussians retained after compression; for example, $r$=0.20 means 20% of the original Gaussians are preserved. For each scene we show (left to right): compressed input rendering, our reconstruction, and zoom-in patches. Our method preserves structure while recovering fine textures.

*UGC* preserves structural fidelity while *GaussPainter* effectively restores texture and color consistency, yielding perceptually sharper details and fewer artifacts.

**Outdoor benchmarks at 3% and 5% ratios.** Figure 10 shows additional results on Mip-NeRF360 at extreme ratios $r$=0.03 and $r$=0.05. Despite aggressive compression, the combination of mask-guided conditioning and one-step diffusion maintains coherent geometry and suppresses color drift in low-texture regions, demonstrating robustness in challenging outdoor scenes.

## A.4 DETAILS OF DATA SOURCES AND TRAINING DETAILS

The Gaussian scenes used in our experiments are derived from publicly available datasets. For indoor benchmarks, ScanNet++ and Replica Gaussian scenes are obtained from the SceneSplat repository (Li et al., 2025). For outdoor scenes on Mip-NeRF360 (Barron et al., 2021) and DL3DV (Ling et al., 2024), Gaussian scenes are generated using the 3DGS pipeline with 30,000 optimization steps; densification and other hyperparameters follow the default 3DGS settings unless otherwise stated.

We train our models using two data sources: DL3DV and ScanNet++. The training set includes all video sequences from DL3DV and the official training split of ScanNet++. For validation, we randomly render 100 novel views from the training scenes. For testing, we use the 50 official test scenes from ScanNet++ and all scenes from Replica and Mip-NeRF360.

## A.5 GENERALIZATION TO UNSEEN SCENES AND PERFORMANCE TRENDS.

To further evaluate how well our method generalizes beyond the training distribution, we conduct experiments on two scenes from the Tanks & Temples dataset, the Mip-NeRF360 dataset, and the Replica dataset. All these scenes are not included in our training data, ensuring that the model has never observed any images from these environments, and thus Tanks & Temples, Mip-NeRF360, and Replica all serve as out-of-domain data for testing generalization performance.

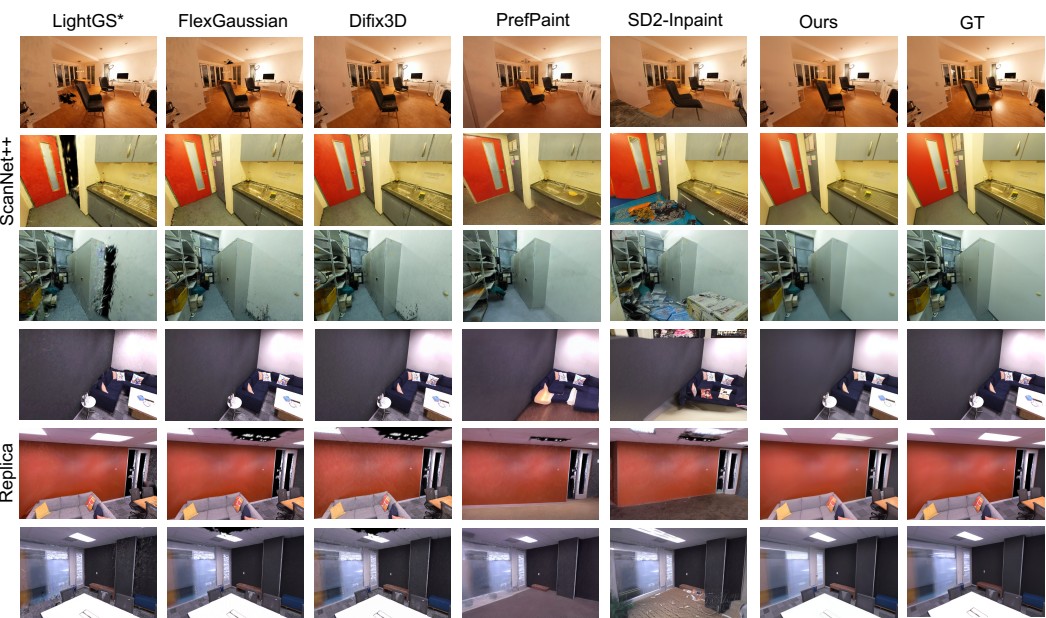

Figure 8: **Indoor results at** $r$=0.30 **(30% pruning).** Additional qualitative comparisons on Scan-Net++ and Replica. Here $r$=0.30 means that 30% of the original Gaussians are retained. Even under this stronger compression, the proposed pipeline mitigates artifacts and maintains perceptual quality.

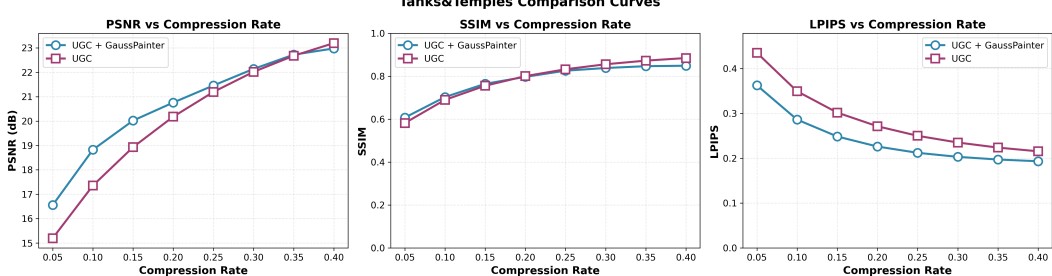

Figure 9: **Outdoor results on Tanks & Temples datasets** This figure presents three curves (PSNR, SSIM, LPIPS) of UGC + GaussPainter and UGC under different compression rates on the unseen scenes of Tanks & Temples dataset. Across metrics, UGC + GaussPainter performs better or comparable to UGC in most cases—its gains are prominent under aggressive compression.

The figure 9 reports the PSNR, SSIM, and LPIPS curves under different compression rates. Across the three metrics, UGC + GaussPainter performs consistently better or comparable to UGC across most compression levels. The gains are most notable under aggressive compression, where the Gaussian representation becomes extremely sparse and GaussPainter effectively recovers missing structures. At higher compression budgets, UGC alone occasionally achieves slightly better scores; we believe this is because the Tanks & Temples scenes contain complex outdoor geometry and fine textures, where the generative refinement may introduce subtle artifacts when the underlying Gaussian representation is already of high quality. As summarized in Table 2, the Replica results are reported in this table. The quantitative comparisons on the Mip-NeRF360 dataset are reported in Table 3.

Overall, these results show that our approach generalizes well to unseen scenes and provides clear benefits under challenging compression settings, while maintaining competitive performance in high-quality regimes.

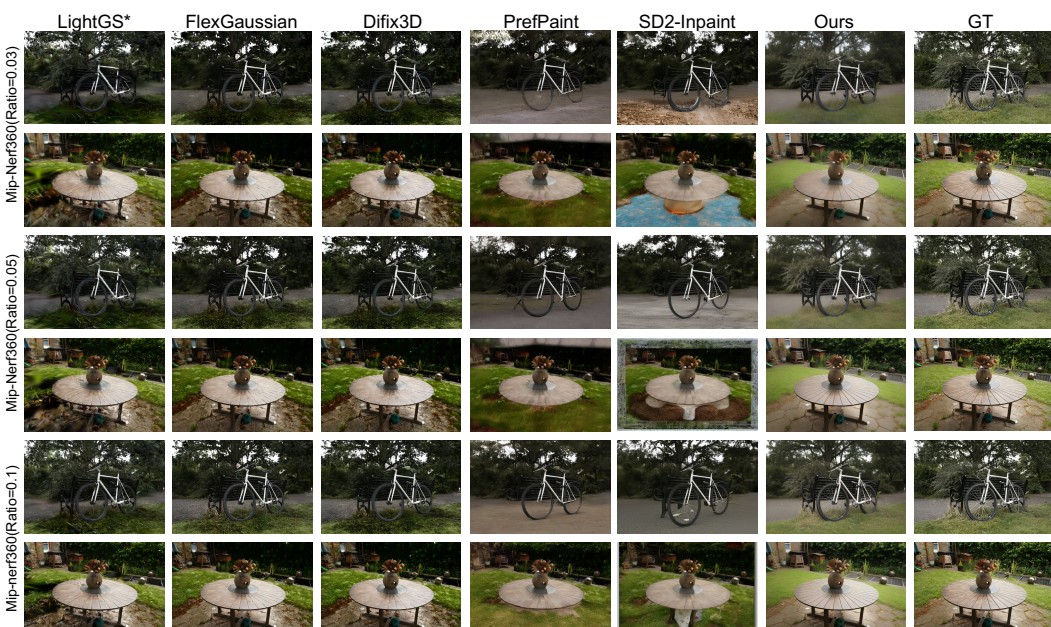

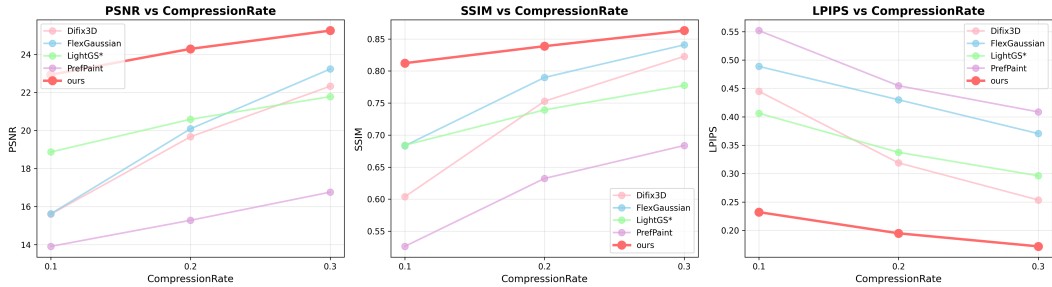

Figure 10: **Outdoor results on Mip-NeRF360 at** $r$=0.03 **and** $r$=0.05. Here $r$=0.03 and $r$=0.05 indicate that only 3% and 5% of the original Gaussians are retained, respectively. Despite such extreme compression, our approach preserves coherent geometry and reduces color drift, particularly in low-texture or large-hole regions.

Figure 11: **Quantitative results on ScanNet++ under different compression ratios of Gaussian counts, reporting LPIPS↓, SSIM↑, PSNR↑.**

LLM USAGE

We used OpenAI's ChatGPT strictly as a general-purpose assistive tool for language and formatting. Specifically, it was employed for text refinement (grammar and clarity), LaTeX formatting and macro fixes (e.g., equation/table reflow and citation consistency), and figure layout suggestions. The model did not contribute to research ideation, methodology, experimental design, data collection, analysis, or interpretation, nor did it generate novel technical content or claims. All conceptual ideas and final decisions were made by the authors.

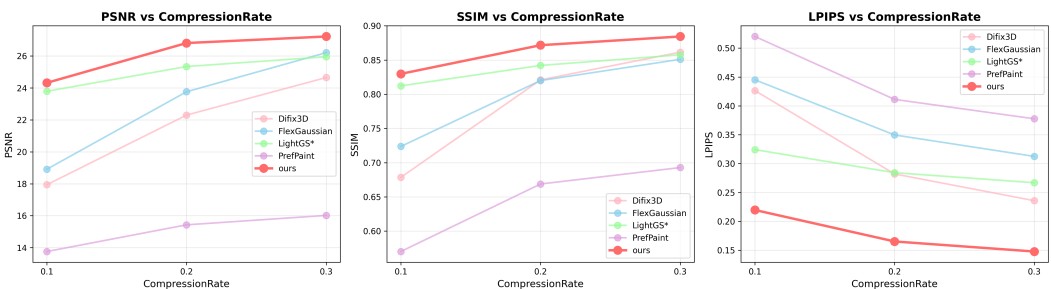

Figure 12: **Quantitative results on Replica under different compression ratios of Gaussian counts, reporting LPIPS↓, SSIM↑, PSNR↑.**

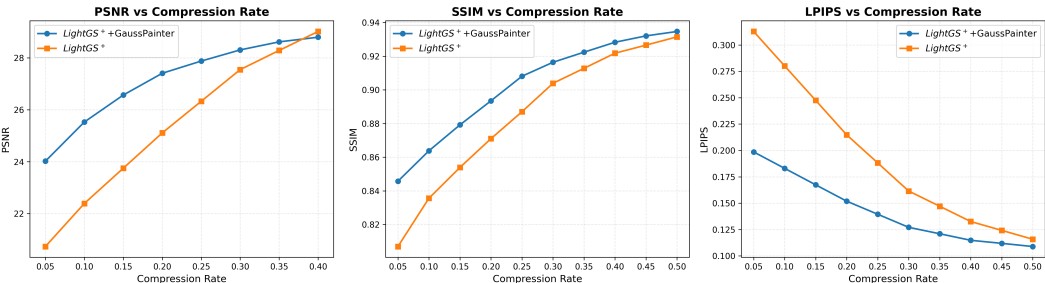

Figure 13: **Quantitative results on ScanNet++ under different compression ratios of Gaussian counts, comparing the performance of LightGS and LightGS+GaussPainter across PSNR, SSIM, and LPIPS metrics.**