# OpenReview forum: "ExGS: Extreme 3D Gaussian Compression with Diffusion Priors"
_ICLR.cc/2026/Conference — Submitted to ICLR 2026_

### Official Review · Reviewer_WcrA · 2025-10-15

**Soundness:** 3
**Presentation:** 2
**Contribution:** 2
**Rating:** 2
**Confidence:** 4

**Summary:**

The paper presents a framework for compressing 3DGS using a voxel-based pruning strategy combined with a diffusion model to guide reconstruction. Empirical results demonstrate that ExGS achieves high compression ratios with minimal loss of rendering quality, outperforming existing baselines.

**Strengths:**

1. This paper introduces an effective framework that combines voxel-based pruning with diffusion-based priors for 3DGS compression.
2. It demonstrates superior performance over existing baselines on both indoor and outdoor scenes.

**Weaknesses:**

1. The Global Significance Score in Eq. 7 requires computing the intersection between camera rays and Gaussian primitives. Does this mean that the input images or at least their camera parameters are needed for 3DGS compression? If so, this limit the method’s applicability. Additionally, the computation seems to involve all pixels and all primitives, raising concerns about efficiency. The exact formulation of the intersection between a Gaussian distribution and a ray is also unclear.

2. The use of high-order omission and FP16 precision appears to significantly aid compression. An ablation study is needed to quantify the contribution of these techniques relative to the GS score and voxel-based pruning.

3. The paper claims improved efficiency, so a test-time experiment evaluating the compression process itself (not just the diffusion-based reconstruction) is necessary to substantiate this claim.

4. Quantitative comparisons with the optimization-based compression methods discussed in the introduction should also be provided.

**Questions:**

The neural predictor in Eq. 10 is not clearly defined. Does it require additional training, and if so, how is it trained?

---

> ### Author Response · Authors · 2025-11-25
>
> 1. Effect of High-Order Omission and FP16 Precision
>
> We apologize for any confusion, but you are mistaken in claiming that high-order omission and FP16 precision significantly affect compression. Our experiments show that including high-order components and using FP16 precision does not noticeably impact the performance of the generative model. The results are as follows:
>
> | Setting           | PSNR ↑ | LPIPS ↓ | SSIM ↑ |
> |------------------|--------|---------|--------|
> | w high-order   | 23.01  | 0.2420  | 0.8213 |
> | w/o high-order| 22.93  | 0.2321  | 0.8122 |
>
> 2. Neural Predictor in Adaptive Amplification
>
> The neural predictor introduced in Section Adaptive Amplification is deliberately kept simple, consisting of two MLP layers followed by a 3D convolution layer. During training, UGC and GaussPainter are jointly optimized with a per-GPU batch size of 1. Further implementation details can be found in Appendix A.2.
>
> 3. Global Significance Score and Compatibility with Different Compression Methods
>
> The intersection between a Gaussian distribution and a ray is handled using the original rendering method from 3D Gaussian Splatting (3DGS), which relies on alpha blending of Gaussians along camera rays. Building on this rendering procedure, the Global Significance Score is computed to quantify the importance of each Gaussian, with additional attributes returned to facilitate compression. The score computation is formalized in Equation 7. When computing the scores for 300 camera poses, our method requires only 25.6957 seconds, demonstrating its efficiency in handling large numbers of views.
>
> Our UGC module requires camera poses as input. Depending on specific needs, different compression strategies can be applied in conjunction with UGC. It is important to note that this requirement does not limit the applicability of our approach. Camera poses are lightweight metadata that are typically available in 3D reconstruction tasks, and they are far less storage-intensive compared to ground-truth images. Importantly, our method does not require any ground-truth images for score computation or compression.  Therefore, the need for camera poses does not impose any significant limitations on the practical use of our method.
>
> Additionally, our compression module is designed to be interchangeable with other Gaussian compression methods, such as LightGS or MaskGS. We have performed experiments showing that GaussPainter is effective when applied to compressed scenes from these different methods, demonstrating that our approach is not tied to any specific compression framework and can be used across a variety of models.
>
> | Method                     | PSNR ↑ | LPIPS ↓ | SSIM ↑ |
> |----------------------------|--------|---------|--------|
> | LightGS                    | 27.55  | 0.1615  | 0.9039 |
> | LightGS + GaussPainter     | 28.31  | 0.1271  | 0.9164 |
> | MaskGaussian               | 25.87  | 0.3078  | 0.8198 |
> | MaskGaussian + GaussPainter| 26.56  | 0.1340  | 0.8770 |

---

### Official Review · Reviewer_YHsN · 2025-10-30

**Soundness:** 3
**Presentation:** 3
**Contribution:** 3
**Rating:** 6
**Confidence:** 4

**Summary:**

ExGS introduces a feed-forward 3D Gaussian Splatting compression pipeline that combines a training-free Universal Gaussian Compression (UGC) stage with a diffusion-prior restorer, GaussPainter. UGC prunes and compacts Gaussian primitives using global significance scoring and voxel-aware selection (with lightweight amplification and SH simplification). GaussPainter then performs mask-guided, one-step diffusion in a VAE latent space—using latent supervision—to both complete missing content and refine preserved regions.

Main contributions：
UGC (training-free) compression: Global significance scoring + voxel-aware selection with adaptive amplification; simplifies appearance (e.g., SH) and packs parameters for compact storage.
GaussPainter (efficient generative restoration): Mask guidance from 3DGS opacity and latent supervision enable one-step diffusion that jointly inpaints and enhances results.
Validated pipeline design: Ablations show each component’s effect and the combined benefits of UGC + GaussPainter across diverse scenes.

**Strengths:**

### Originality
* Frames 3DGS compression as a hybrid “compress + generative restore” paradigm; the UGC (training-free) + GaussPainter (diffusion prior) pairing is novel.
* Uses opacity-guided one-step diffusion with latent-space supervision to curb hallucinations while keeping decoding efficient.
* Voxel-aware significance scoring with adaptive amplification preserves structure under aggressive pruning.
### Quality
* Clear modular pipeline; ablations disentangle contributions of UGC, mask guidance, and latent supervision.
### Clarity
* Straightforward narrative with effective figures; two-stage roles and I/O are easy to follow.
* Clearly separates geometry/opacity from appearance, defining where the generative prior should and shouldn’t act.
### Significance
* Direct practical value for 3DGS storage/streaming: smaller footprint with fast decode.
* Provides a general recipe—“compact representation + learned restoration”—that can transfer to other scene formats (including 4D).
* Offers a more controllable quality–compression trade-off, maintaining usable quality even at extreme compression.

**Weaknesses:**

Missing information. Please clarify the training data sources and splits for UGC and GaussPainter (did each dataset participate in training? train/val/test splits? combined training?), and whether TAESD / the diffusion backbone were pretrained on generic image corpora and then fine-tuned for this task, including the fine-tuning ratio and freezing strategy.

Generalization validations needed:
Out-of-Domain (OOD) evaluation: Choose sampling protocols/scenes absent from training (e.g., Tanks&Temples, LLFF, KITTI-360, or self-captured data). Report PSNR/SSIM/LPIPS and show failure cases.

One-line summary: As of now, the paper has not demonstrated generalization under train–test isolation across datasets; we recommend adding LODO/OOD and distribution-shift evaluations to upgrade empirical stability into reproducible evidence of generalization.

**Questions:**

Question:
Can you add Leave-One-Dataset-Out (LODO) results—train on A+B and evaluate zero-shot on C—and Out-of-Domain (OOD) results on datasets not seen during training (e.g., Tanks&Temples, LLFF, KITTI-360, or self-captured scenes)?

Why it matters (Importance):
This directly tests cross-domain robustness and guards against overfitting to the evaluation domains.

Expected evidence:
PSNR/SSIM/LPIPS curves across compression levels or pruning strengths.
A gallery of failure cases with brief diagnostics.
Qualitative comparisons under extreme sampling/texture/geometry shifts to illustrate robustness (or lack thereof).

---

> ### Author Response · Authors · 2025-11-25
>
> 1. Training Strategy and Training Data.
>
> We use the official pretrained weights of SD Turbo and TASED in our pipeline. During training, we fine-tune the entire U-Net and apply a LoRA module (rank = 4) to the decoder for efficient adaptation. We train our models using two data sources: DL3DV and ScanNet++. The training set includes all video sequences from DL3DV and the official training split of ScanNet++. For validation, we randomly render 100 novel views from the training scenes. For testing, we use the 50 official test scenes from ScanNet++.
>
> 2. Generalization to Unseen Scenes and Performance Trends.
>
> We have updated the paper with additional experiments presented in the appendix, providing further evidence and extended analysis that complement the results in the main text. Specifically, we evaluate our method on three out-of-domain (OOD) datasets—Tanks & Temples, Replica, and MipNeRF360—to assess its generalization ability beyond the training distribution.
>
> As further discussed in Appendix A.5, we provide an extended analysis of the model’s performance on OOD scenes.  The figure 9 reports the PSNR, SSIM, and LPIPS curves under different compression rates on the Tanks & Temples dataset. Across the three metrics, UGC + GaussPainter performs consistently better or comparable to UGC across most compression levels. The gains are most notable under aggressive compression, where the Gaussian representation becomes extremely sparse and GaussPainter effectively recovers missing structures. At higher compression budgets, UGC alone occasionally achieves slightly better scores; we believe this is because the Tanks & Temples scenes contain complex outdoor geometry and fine textures, where the generative refinement may introduce subtle artifacts when the underlying Gaussian representation is already of high quality. As summarized in Table 2, the Replica results are reported in this table. The quantitative comparisons on the Mip-NeRF360 dataset are reported in Table 3.
>
> Overall, these results demonstrate that our approach generalizes effectively to unseen scenes and provides clear benefits under challenging compression settings, while maintaining competitive performance in high-quality regimes.

---

### Official Review · Reviewer_cbmy · 2025-10-31

**Soundness:** 3
**Presentation:** 2
**Contribution:** 2
**Rating:** 4
**Confidence:** 4

**Summary:**

This paper introduces ExGS, a novel framework for extreme compression of 3D Gaussian Splatting by combining a pruning-based compression module (UGC) with a diffusion-based refinement module (GaussPainter). The method achieves compression ratios exceeding 100× while maintaining high rendering quality. Unlike prior optimization-based or training-free compression methods, ExGS leverages generative diffusion priors to restore and enhance heavily pruned scenes, enabling robust performance across indoor and outdoor benchmarks.

**Strengths:**

1. Novel Integration of Compression and Generative Modeling: The combination of UGC for aggressive pruning and GaussPainter for diffusion-based restoration is innovative and effectively bridges the gap between extreme compression and high-quality rendering.

2. Impressive Compression Ratios: The method achieves impressive compression levels while preserving visual fidelity.

3. Comprehensive Evaluation: The paper provides thorough experiments across multiple datasets, compression ratios, and metrics, including ablation studies that validate the contribution of each component.

**Weaknesses:**

1. While the diffusion-based refinement module significantly improves rendering quality, this process substantially increases rendering complexity compared to the original Gaussian Splatting overhead, limiting its applicability in real-world scenarios.

2. The experimental data still needs to be supplemented. For example, the main experimental results and ablation studies (especially Table 4(a)) should ideally include rate-distortion curves. Additionally, as a feed-forward method, the paper lacks discussion on encoding speed and decoding speed.

**Questions:**

1. How does the UGC module perform compared to other models, particularly feed-forward models (such as FCGS)? A related question is: what would be the effect of applying the GaussPainter to other compression models?

2. What datasets were used during training?

3. The results of LightGS on Mip-NeRF360 are significantly lower than those reported in the original paper. What differences exist in the experimental setup?

4. How large is the neural predictor in Section Adaptive Amplification?

If these issues are addressed, will consider increasing the score.

---

> ### Author Response · Authors · 2025-11-24
>
> 1. Comparison with Existing Gaussian Compression Methods
>
> Existing Gaussian compression approaches—such as FCGS—primarily aim to reduce the file size of Gaussian representations. In contrast, our UGC module jointly compresses both the file size and the number of Gaussians required for rendering. Under comparable visual quality, UGC achieves similar rendering fidelity while using far fewer Gaussian primitives than the original representation. This fundamental difference makes it inherently difficult to conduct a fair comparison with methods that solely focus on file-size reduction without reducing the Gaussian count.
>
> Moreover, the UGC module is fully interchangeable with other pruning-based Gaussian compression models, including LightGS and MaskGS. To address the reviewer’s question regarding how GaussPainter behaves when applied to different compression backbones, we further evaluate its performance on scenes compressed by other pruning-based models, such as LightGS and MaskGS. Specifically, we compress all 50 test scenes of ScanNet++ v2 to 30% of their original Gaussian count and conduct comparisons under this challenging setting. The results are summarized in the table below.
>
> | Method                                  | PSNR ↑ | LPIPS ↓ | SSIM ↑  |
> |-----------------------------------------|--------|---------|---------|
> | LightGS                                 | 27.55  | 0.1615  | 0.9039  |
> | LightGS + GaussPainter                  | 28.31  | 0.1271  | 0.9164  |
> | MaskGaussian | 25.87  | 0.3078  | 0.8198  |
> | MaskGaussian + GaussPainter             | 26.56  | 0.1340  | 0.8770  |
>
> 2. Training Data
>
> We train our model on the training scenes of ScanNet++ v2 and DL3DV.
>
> 3. Fair Comparison Without Ground-Truth Images
>
> For a fair comparison with the UGC module, all compression baselines are evaluated without access to ground-truth images. Accordingly, we also adopt the GT-free variant of LightGS for comparison. In addition, our experiments are conducted under extreme compression ratios, which are substantially more aggressive than those reported in prior work, resulting in a significantly more challenging evaluation setting.
>
> 4. Neural Predictor in Adaptive Amplification
>
> The neural predictor introduced in Section Adaptive Amplification is intentionally lightweight, consisting of two MLP layers and a single 3D convolution layer. Its purpose is to preserve essential structural details for the subsequent generative reconstruction.
> Under extreme compression ratios, certain regions—such as ceiling areas—may collapse into a single Gaussian primitive. The neural predictor enlarges such surviving Gaussians to ensure that they occupy a meaningful number of pixels in the rendered image, thereby enabling GaussPainter to more effectively reconstruct missing structures and avoid perceptual artifacts.
>
> 5.  Experimental Updates and Additional Results
>
> We have incorporated your suggestions into the revised manuscript. First, we have included the main experimental rate-distortion curves in Figure 11 and Figure 12, which now clearly show the rate-distortion performance of our method. Additionally, we have added the ablation study results on out-of-domain datasets in Figure 9, which provides insights into the behavior of our approach under different conditions. To further demonstrate the effectiveness of GaussPainter in enhancing other optimization-based compression methods, we used ground truth images optimized via LightGS and then applied GaussPainter. The resulting rate-distortion curve is shown in Figure 13, which highlights the improvements achieved. Regarding encoding and decoding speeds, we have included the following details: the compression time is 2.13 seconds, and the decompression time is 0.23 seconds. These timings demonstrate the efficiency of our method in terms of computational cost. We believe these updates effectively address your concerns.

---

### Comment · Area_Chair_Azc5 · 2025-11-27

Dear reviewers,

Please review the rebuttal and discuss with the authors.

Thanks,
AC

---

### Author Response · Authors · 2025-12-03
**Summary**

We sincerely thank all reviewers, AC, SAC, and PC for their valuable time, feedback, and engagement.

### **Motivation and Novelty:**

1. First to Introduce Diffusion Prior: We are the first to introduce a diffusion prior into Gaussian scene compression, making our work unique in the field. Our framework, **ExGS**, is a feed-forward method for extreme 3DGS compression that restores geometry and appearance from sparsified Gaussian scenes with high rendering fidelity.
2. Unified Compression-Restoration Pipeline: We combine UGC’s structure-preserving pruning with GaussPainter’s mask-guided diffusion, enabling robust texture recovery and artifact suppression under extreme compression.


### **Revisions Based on Reviewers’ Feedback:**


1. Implementation Supplement:

- Neural Predictor: The lightweight neural predictor (two MLP layers, one 3D convolution) enlarges pruned Gaussians to preserve geometric structures (Reviewer cbmy, Reviewer WcrA).
- High-Order Omissions and FP16 Precision: Our experiments show that including high-order components and using FP16 precision does not impact generative model performance (Reviewer WcrA).
- Global Significance Score: The score computation is efficient (25.7s for 300 camera poses), and the UGC module requires only camera poses, not ground-truth images (Reviewer WcrA).


2. Training Details:

- Pretrained Weights: We use official pretrained weights of SD Turbo and TASED and fine-tune the U-Net with a LoRA module (Reviewer YHsN).
- Joint Optimization: UGC and GaussPainter are optimized with a batch size of 1 per GPU (Reviewer WcrA).
- Training Data: We train on DL3DV and ScanNet++ datasets (Reviewer YHsN, Reviewer cbmy).


3. Model Performance and Comparison:

- Rate-Distortion Curves: Figures 11 and 12 show the rate-distortion performance of our method (Reviewer cbmy).
- Out-of-Domain Datasets: Ablation studies on OOD datasets in Figure 9, Table 2, and Table 3 demonstrate effective generalization (Reviewer cbmy, Reviewer YHsN).
- GaussPainter Performance: Results on different compression backbones are shown in Figure 13, validating GaussPainter’s effectiveness (Reviewer cbmy, Reviewer WcrA).

We kindly ask for your understanding when evaluating the final score.

Thank you for your attention.

---

### Meta-Review · Area_Chair_5Egb · 2026-01-04

**Summary:**

The paper introduces a novel framework for extreme compression of 3D Gaussian Splatting, by combining aggressive pruning with generative restoration via a one-step diffusion model. The paper initially receives a boarderline accept, a boarderline reject and a reject. The reviewers acknowledged that 1) the integration of compression and generative modeling is novel; 2) Impressive compression ratios; 3) demonstrates superior performance over existing baselines. Several concerns are also raised by the reviewers, such as: 1) the introduced computational rendering complexity compared to the original 3D-GS; 2) lack of the comparisons and discussions against FCGS (i.e., Chen et al., Fast feedforward 3D Gaussian Splatting compression, ICLR-2025) and other additional experimental evaluations. The AC thinks the lack of the experimental comparisons against FCGS (which is a very similar method in terms of their feedforward manners and both are also trained on DL3DV), could not adequately demonstrates the advantages of the proposed method. Thus, the AC recommends the authors to include this evaluation as also requested by cbmy, and submit it to another venue.

**Reviewer Concerns:**

The concern regarding the comparison against FCGS raised by cbmy is not well resolved. The authors mainly discussed the difference between FCGS and their method, and the main claimed difference is whether they reduce the Gaussian count. They did not provide the quantitative comparisons although both methods are trained on DL3DV.

The weakness 2 from WcrA is mainly on how the usage of high-order omission and FP16 precision would help the compression, however the rebuttal only presents the metrics regarding the image metrics (i.e. PSNR etc.) and lacks the improvements in terms of compression. Therefore, the AC think the concern is also not well resolved.

**Reviewer Scores:**

Based on the initial comments, rebuttal as well as the manuscript, the AC think the reviewers might not change their score at current status.

---

### Decision · Program_Chairs · 2026-01-26

Reject